# Breast Cancer Mortality among Women with a *BRCA1* or *BRCA2* Mutation in a Magnetic Resonance Imaging Plus Mammography Screening Program

**DOI:** 10.3390/cancers12113479

**Published:** 2020-11-23

**Authors:** Ellen Warner, Siqi Zhu, Donald B. Plewes, Kimberley Hill, Elizabeth A. Ramsay, Petrina A. Causer, Jean Seely, Roberta A. Jong, Pamela Lenkov, Christine Elser, Pavel Crystal, Martin J. Yaffe, Vasily Giannakeas, Ping Sun, Steven A. Narod

**Affiliations:** 1Odette Cancer Centre, Sunnybrook Health Sciences Centre, Toronto, ON M4N 3M5, Canada; siqi.zhu@sunnybrook.ca (S.Z.); Kimberly.Hill@grhosp.on.ca (K.H.); eramsay3009@gmail.com (E.A.R.); Petrina.Causer@nygh.on.ca (P.A.C.); Roberta.Jong@sunnybrook.ca (R.A.J.); Pamela.Lenkov@sunnybrook.ca (P.L.); martin.yaffe@sri.utoronto.ca (M.J.Y.); 2Department of Medical Biophysics, University of Toronto, Toronto, ON M5G 1L7, Canada; don.plewes@gmail.com; 3Department of Radiology, University of Ottawa, Ottawa, ON K1N 6N5, Canada; jeseely@toh.ca; 4Marvelle Koffler Breast Centre, Mt. Sinai Hospital, Toronto, ON M5G 1X5, Canada; Christine.Elser@sinaihealth.ca; 5Department of Medical Imaging, University Health Network, Toronto, ON M5T 1W7, Canada; damselindistress613@gmail.com; 6Women’s College Research Institute, University of Toronto, Toronto, ON M5G 1N8, Canada; vasily.giannakeas@wchospital.ca (V.G.); Ping.Sun@wchospital.ca (P.S.); Steven.Narod@wchospital.ca (S.A.N.)

**Keywords:** breast neoplasms, *BRCA1* gene, *BRCA2* gene, magnetic resonance imaging, screening, mortality

## Abstract

**Simple Summary:**

Women with a *BRCA1* or *BRCA2* gene mutation have up to an 80% lifetime risk of breast cancer unless their breasts are surgically removed, but many decline or defer surgery and choose screening, hoping that if cancer occurs, it will be detected at a curable stage. In this study 489 women with a *BRCA1 or BRCA2* mutation aged from 25 to 65 years, who had never had breast or ovarian cancer, were screened annually with breast magnetic resonance imaging (MRI) in addition to mammography and were followed for an average of 13 years (range: 9 to 23 years). Ninety-five of the 489 women enrolled in the study had a bilateral preventive mastectomy in the follow-up period. Of the 91 women diagnosed with breast cancer, four died of breast cancer. The most common cause of death was ovarian cancer. For women with *BRCA* mutations who choose annual screening with MRI and mammography, the probability of dying of breast cancer within 20 years is 2%.

**Abstract:**

Annual breast magnetic resonance imaging (MRI) plus mammography is the standard of care for screening women with inherited *BRCA1/2* mutations. However, long-term breast cancer-related mortality with screening is unknown. Between 1997 and June 2011, 489 previously unaffected *BRCA1/2* mutation carriers aged 25 to 65 years were screened with annual MRI plus mammography on our study. Thereafter, participants were eligible to continue MRI screening through the high-risk Ontario Breast Screening Program. In 2019, our data were linked to the Ontario Cancer Registry of Cancer Care Ontario to identify all incident cancers, vital status and causes of death. Observed breast cancer mortality was compared to expected mortality for age-matched women in the general population. There were 91 women diagnosed with breast cancer (72 invasive and 19 ductal carcinoma in situ (DCIS)) with median follow-up 7.4 (range: 0.1 to 19.2) years. Four deaths from breast cancer were observed, compared to 2.0 deaths expected (standardized mortality ratio (SMR) 2.0, *p* = 0.14). For the 489 women in the study, the probability of not dying of breast cancer at 20 years from the date of the first MRI was 98.2%. Annual screening with MRI plus mammography is a reasonable option for women who decline or defer risk-reducing mastectomy.

## 1. Introduction

One in 400 women in the general population carry a germline mutation in *BRCA1* or *BRCA2.* For these women, the estimated risk of developing breast cancer by age 70 ranges from 45% to 88% [1,2,3]. Bilateral risk-reducing mastectomy with or without reconstruction is currently the most effective strategy for reducing breast cancer risk mortality. The residual risk of breast cancer after this surgery is only 1% to 2% [4]. However, many women forego preventive surgery for a variety of reasons. Developing cancer is not a certainty; risk-reducing salpingo-oophorectomy and antiestrogen therapy may reduce breast cancer incidence, particularly for *BRCA2* mutation carriers [5]. For some, the potential effects of a mastectomy on body image or sexual function (even with state-of-the-art reconstruction) may be unacceptable [6]. Some women choose to defer this surgery until they have established a secure intimate relationship or have completed childbearing and breastfeeding [6,7].

Annual screening with contrast-enhanced magnetic resonance imaging (MRI) in addition to mammography, starting at age 25 or 30, is now the standard of care for female mutation carriers who decline or defer bilateral risk-reducing mastectomy [8,9,10,11]. The sensitivity of this screening regimen exceeds 95%, with 85% of cancers detected at an early stage (smaller than 2 cm with no nodal involvement) [12]. However, little is known about long-term breast cancer-specific mortality for women who opt for annual breast screening with MRI. This information is critical for enabling health care providers to help *BRCA* mutation carriers make informed decisions about management of their risk.

In 1997, we began a prospective observational study of *BRCA* mutation carriers who were screened annually with MRI and mammography. Women had not been diagnosed with breast or ovarian cancer prior to enrollment. We report the long-term breast cancer-specific survival and causes of death for the 489 women in the cohort.

## 2. Results

Of the 502 unaffected women enrolled in the study, 13 were excluded because they did not complete the first MRI. Of the remaining 489 women, 248 had a *BRCA1* mutation and 241 had a *BRCA2* mutation. The characteristics of the study population are presented in Table 1.

The median duration of follow-up from the first MRI examination for the cohort was 12.7 years (range: 8.5–22.5 years). Of the 489 women, 71.2% had ten or more years of follow-up. Ninety-five of the 489 women underwent a preventive bilateral mastectomy in the follow-up period. The mean age of preventive mastectomy was 43.4 years (range: 28 to 58 years). The mean time from first MRI to preventive mastectomy was 4.3 years (range: 0.01 to 20.1 years). There were four cases of breast cancer diagnosed at preventive mastectomy. All four of these women are currently alive. Among the 95 women who had preventive mastectomy, there were no cases of breast cancer diagnosed after the surgery.

A total of 91 breast cancers were diagnosed, including 72 invasive cancers and 19 cases of ductal carcinoma in situ (DCIS). Among the 91 women, 60 cases of breast cancer were diagnosed prior to December 2011 and 31 were diagnosed from 2012 to 2016. The characteristics of the breast cancers are presented in Table 2. Of the 66 invasive cancers with stage details available, 49 (74%) were node-negative and less than or equal to 2 cm in size (stage I).

Of the 60 cancers diagnosed during the study period, 28 were detected by MRI alone, 19 were detected by MRI and another screening method, one was detected by mammography alone, one was detected by ultrasound alone, three were palpable, one was an interval cancer and four were discovered at prophylactic mastectomy (for three women, the means of diagnosis was missing). For the 31 cancers detected in the follow-up period, the mode of detection is unknown. Given that most women would have enrolled in the provincial high-risk screening program, most would likely have been detected by MRI [12].

The annual risk of breast cancer was 2.3%. The risk was 2.0% for *BRCA1* mutation carriers and 2.6% for *BRCA2* mutation carriers. The ten-year cumulative incidence of breast cancer for the entire cohort was 20.6% and the 15-year cumulative incidence of breast cancer was 24.1% (Figure 1). 

Of the 489 women who were in the cohort, 95 had a bilateral preventive mastectomy in the follow-up period. All four of the deaths were among the 394 women who did not have a bilateral mastectomy. Given that there were no deaths from breast cancer following a bilateral mastectomy, it was not possible to estimate the hazard ratio for death from breast cancer for women associated with bilateral preventive mastectomy.

There were four deaths attributed to breast cancer among the 73 patients with invasive breast cancer, as summarized in Table 3. 

Patient #1 (296) was a *BRCA2* mutation carrier who was diagnosed on her fifth round of screening at age 63 while taking tamoxifen for chemoprevention. The tumor was a 0.7-cm, grade 2, node-negative, ER-positive, PR-negative, invasive ductal carcinoma. She declined adjuvant hormonal therapy and developed distant metastases seven years later. She died at age 74. 

Patient #2 (424) was a *BRCA1* mutation carrier, who delivered a baby 13 months after her last screening MRI. Twenty-four months post-delivery at age 33 she was diagnosed with a (self-detected) invasive ductal carcinoma, with 28 of 31 lymph nodes positive. She died of distant metastases at age 37.

Patient #3 (455) was a *BRCA1* mutation carrier diagnosed with breast cancer on her first screening with MRI at age 48. The cancer was 3-cm, ER-positive and node-positive. She received adjuvant chemotherapy and hormonal therapy. She developed diffuse distant metastases six years later and died at age 55. 

Patient #4 (566) was a *BRCA1* mutation carrier diagnosed in 2010 at age 45 at her fifth round of screening with MRI screen-detected cancer. This 1.1-mm, grade I, triple-negative, node-negative invasive ductal carcinoma of the breast was treated with breast conserving therapy. In July 2015, she had targeted radiation to a solitary liver metastasis with a different molecular profile than the breast primary (GATA-3-negative vs. -positive, respectively). In December 2016, there was liver progression and spread to the peritoneum and lung was noted. Following one cycle of carboplatin, paclitaxel +/− veliparib, she had a lower GI bleed, which led to the diagnosis of a cancer of likely intestinal origin (GATA-3-negative). Whether this was a primary or metastatic lesion could not be determined. The peritoneal disease responded to the single-course chemotherapy but there was liver progression. A second liver biopsy then showed GATA-3-positive disease. The patient died in July 2018 at age 53. 

For the 72 women with invasive breast cancer, the median follow-up was 7.4 (range: 0.1 to 19.2) years. The ten-year actuarial survival was 94.0% and the twenty-year actuarial survival was 90.1% (Figure 2). Overall, for the 489 women in the study, the actuarial risk of not dying of breast cancer at 20 years from the date of the first MRI screening was 98.2%. Among the 394 women who did not have preventive mastectomy, the actuarial risk of not dying of breast cancer at 20 years was 98.1%.

There were four deaths from breast cancers observed in the cohort. Based on SEER rates of breast cancer incidence and breast cancer survival, there were 2.0 deaths expected (standardized mortality ratio (SMR) 2.0) and this difference was not statistically significant (*p* = 0.14). (It is not clear if the fourth case was a death from breast cancer; if we exclude her, the SMR was 1.5).

There were 10 additional deaths from cancer (five from ovarian/tubal/peritoneal cancer, four from pancreatic cancer and one from lung cancer) and four deaths from other causes.

## 3. Discussion

In this study, we followed 491 women with a germline *BRCA* mutation, who were annually screened with MRI and mammography for a median of 12.7 years. The incidence of breast cancer was 2% per year. There were 19 cases of DCIS diagnosed and none of these women died of breast cancer. There were 72 cases of invasive cancer diagnosed, with four breast cancer-related deaths, and for these 72 women, the estimated 20-year survival was 90%, while for the entire cohort of 491 women, the 20-year probability of breast cancer-free survival was 98.2%. The standardized mortality ratio from breast cancer was not significantly different from that of the general population.

These results will be very reassuring to women who are reluctant to undergo risk-reducing mastectomy. Other women will still choose to undergo risk-reducing mastectomy to avoid the inconvenience and stress of annual scans (and sometimes biopsies) and the possibility of a breast cancer diagnosis and treatment (radiation, chemotherapy and/or hormonal therapy) with its side effects and complications. 

A recent concern is the finding of long-term retention of gadolinium in the brain after repeated contrast-enhanced MRI scans, although no neurologic consequences have been reported [13]. However, for women struggling with the decision of breast screening vs. mastectomy, a critical question is the difference in breast cancer-specific mortality between the two management strategies. 

In a recent report from the Hereditary Breast and Ovarian Cancer Netherlands (HEBON) study [14], from a registry of patients with germline *BRCA* mutations identified at eight Dutch academic centers, 2857 previously unaffected female mutation carriers were identified, of whom 1128 opted for risk-reducing mastectomy and 1729 opted for breast surveillance, which consisted of annual MRI and mammography for women aged 25 to 60 years. Median follow-up from enrollment in the registry was 10.3 years. Median follow-up from breast cancer diagnosis was not stated explicitly but was approximately 6.5 years. In the *BRCA1* mutation group, there were 268 breast cancers and 20 breast cancer-related deaths among the 990 women in the surveillance cohort (2.0%) and eight breast cancers and one death among the 722 women in the mastectomy cohort (0.1%). In the *BRCA2* mutation group, there were 144 breast cancers and seven breast cancer-related deaths among the 739 women in the surveillance cohort (0.1%) and no breast cancers in the mastectomy cohort (n = 406). The authors concluded that breast surveillance may be more effective for *BRCA2* than for *BRCA1* mutation carriers. There are several possible explanations for the higher death rate among the *BRCA1* mutation carriers diagnosed with breast cancer in this study compared to ours. Our patients were older at the time of diagnosis than those in the HEBON study (median 49 vs. 45 years, respectively) and the tumor growth rate of *BRCA*-related cancers has been shown to slow progressively with age [15]. In our study, for the first 12 years of our study, all MRI examinations were performed at a single, highly experienced center. Adherence to annual MRI-based surveillance in the HEBON study was not documented. 

There are survival data from other breast MRI surveillance studies. In the Dutch MRI screening (MRISC) study, five of the 51 *BRCA* mutation carriers (10%) with breast cancer developed distant recurrences at a median follow-up of nine years [16]. In the combined UK MARIBS and NICE studies, in which 45 breast cancers were detected, only two women died of breast cancer (both *BRCA1* mutation carriers) after a median follow-up of 12 years [17]. In a recent retrospective study of 42 *BRCA* mutation carriers who opted for annual breast screening with MRI and mammography and were subsequently diagnosed with breast cancer, the 5-year survival after breast cancer was 94% [18].

There are several limitations to our study. Distant breast cancer recurrence was not a measurable outcome for our entire cohort because this is not available in the Ontario Cancer Registry. Nevertheless, through the annual questionnaires completed by all patients diagnosed with breast cancer prior to 2012, none of whom were lost to follow-up, we identified two patients who developed distant metastases and are currently alive. We were unable to document the exact number of MRI and/or mammography examinations that occurred after our active surveillance period was completed in 2011; however, the great majority of participants would have been eligible for the province’s high-risk screening program. It is also possible that some women had a bilateral mastectomy or were diagnosed with breast cancer outside Ontario after the active surveillance was stopped. Finally, our follow-up of the 58% of women who had ER+ breast cancers may not have been long enough, as late distant recurrences of endocrine-sensitive breast cancers are not uncommon [19]. Among the six women who had distant recurrences in our study (including the two still alive with disease), the longest interval between diagnosis and distant recurrence was 7 years. Although Duffy et al. reported that 29% of their *BRCA2* mutation carriers died 10 or more years after diagnosis [20], the interval between distant recurrence and death for ER+ patients is often 5 years or longer [21], so most of their recurrences could have been significantly earlier. 

Two of the four patients diagnosed with breast cancer on our screening study who died of metastatic breast cancer (#1 and #2) likely would have not developed breast cancer if they had undergone risk-reducing mastectomy at the time of their first MRI. However, patient #2 had a 3-year delay of MRI screening due to pregnancy and prolonged lactation, which likely contributed to the poor outcome. Screening guidelines for *BRCA* mutation carriers who choose to breastfeed for more than 6 months now recommend that regular annual screening with MRI and mammography be continued, despite the known reduction in imaging sensitivity in the lactating breast [22]. 

Because with annual MRI and mammography, there is still an interval cancer rate of 5% and about 15% of cancers have axillary nodal involvement at diagnosis [12], some centers recommend staggering MRI and mammography every 6 months. Whether this is more effective than concurrent imaging is currently being studied [10]. More frequent screening with MRI has also been suggested. In a recent study, 137 *BRCA* mutation carriers were screened with MRI every 6 months, plus annual mammography. Fourteen cancers were diagnosed, all node-negative, with no interval cancers. Mammography did not detect any cancers missed by MRI. The number of recalls and biopsies needed to detect one cancer by biannual MRI were 2.8 and 1.7, respectively, in *BRCA1* mutation carriers and 12.0 and 8.0, respectively, in *BRCA2* mutation carriers [23]. 

Breast cancer treatment outcomes for *BRCA* mutation carriers will likely improve as well. The results of a recently completed randomized trial of a PARP inhibitor vs. placebo in addition to standard adjuvant therapy for *BRCA*-related breast cancer are awaited with great interest [24].

In our study, and in the HEBON study [14], the most common cause of death was not breast cancer but death from other malignancies, particularly ovarian cancer. As there is no effective screening for ovarian cancer as of yet, this emphasizes the importance of timely risk-reducing salpingo-oophorectomy performed by a gynecologist experienced in examining the abdomen and pelvis for subclinical malignancy. This surgery should be performed by the age of 40 for *BRCA1* mutation carriers and by the age of 45 for *BRCA2* mutation carriers [9]. Meticulous pathological examination of the ovaries and tubes for early malignancy is also essential, as occult cancers, generally in the fimbriated end of the fallopian tubes, are not uncommon [25]. This surgery has the additional advantage of reducing the risk of breast cancer by up to 50% according to some studies [26,27], although risk reduction was not found in *BRCA1* mutation carriers, nor in *BRCA2* mutation carriers who had undergone this surgery within the previous five years in [28]. Even with meticulous prophylactic surgery and pathological examination, there is still a small risk for primary peritoneal carcinoma [29]. The incidence of ovarian cancer in our patients would undoubtedly have been lower if, at the start of our study in 1997, more had been known about optimal risk reduction management. Furthermore, there has since been significant progress in the treatment of ovarian cancer, particularly the use of PARP inhibitors after the first line of chemotherapy [30]. 

## 4. Materials and Methods

Detailed results of the study population and protocol have been published previously [31,32] and are summarized below.

### 4.1. Study Subjects

Between 1 November 1997 and 30 June 2011, 504 women aged 25 to 65 years with a documented germline *BRCA1* or *BRCA2* mutation, no history of breast or ovarian cancer and two intact breasts were enrolled in a prospective breast screening study. Women with a prior history of other cancers were eligible for inclusion, providing the diagnosis was five or more years in the past. From 1997 until 30 June 2009, screening was performed at the Sunnybrook Health Sciences Centre and then at four additional academic tertiary centers in the province of Ontario, Canada (Women’s College Hospital, Mt. Sinai Hospital, University Health Network (all in Toronto) and The Ottawa Hospital) until 2011. Research ethics board approval was obtained from each institution and all participants provided informed consent (project identification number: 003-2000). Women who became pregnant discontinued screening until four to six months after delivery or weaning. Women diagnosed with breast cancer continued screening unless they underwent bilateral prophylactic mastectomy. For each patient diagnosed with breast cancer in study, method(s) of detection, tumor stage, ER, PR, HER2 and treatments were recorded. For women diagnosed with a second ipsilateral or contralateral breast cancer, only details of the first cancer were recorded for this study. Study subjects were included in the current analysis if they completed at least one successful screening MRI. All study subjects diagnosed with breast cancer prior to 2012 (i.e., during the study enrollment period) completed an annual mailed questionnaire that recorded new diagnoses of breast or other cancers, cancer recurrences and prophylactic mastectomies. 

### 4.2. Screening Protocol

Study participants were screened annually with dynamic gadolinium contrast-enhanced MRI screening during the second week of the menstrual cycle as well as four-view mammography, generally done on the same day as the MRI. Film-screen mammography was performed until 31 December 2007, and digital mammography was performed thereafter. Annual ultrasound screening was also performed initially but was discontinued in 2005. 

In July 2011, MRI became an established screening test in Ontario supported by the Ministry of Health and was administered through the high-risk program of the Ontario Breast Screening Program (OBSP). At that time, MRI screening on a research basis was discontinued. All of the women in this study who did not have preventive surgery were eligible for annual MRI and mammography screening through the OBSP program from 2011 until the present day or until the age of 70. We do not have details of the MRI and/or mammography examinations (date of study or result) for MRI scans done from 2012 onwards, but the results of this program have been published [12].

### 4.3. Record Linkage

In 2019, a record linkage was conducted by linking the identifying information in our study database with that of the Ontario Cancer Registry of Cancer Care Ontario (CCO). For each subject, CCO provided the date of cancer diagnosis, the site of cancer diagnosis, the vital status and the cause of death. CCO recorded incident cases of breast cancer until 31 December 2016 and deaths from cancer until 31 December 2018. It was assumed that subjects who were not listed in the cancer registry as deceased were alive and were residents of Ontario on 31 December 2018. 

### 4.4. Statistical Analysis

For breast cancer incidence, subjects were followed from the date of their first MRI screening until the first of the following: date of diagnosis of breast cancer, date of bilateral preventive mastectomy, date of death or 31 December 2016. The 10- and 15-year actuarial cumulative incidence rates were estimated using the Kaplan–Meir method. 

For breast cancer-specific mortality, subjects were followed from the date of their first MRI until the first of the following: death from breast cancer, date of bilateral risk-reducing mastectomy, death from another cause, date of last follow-up or December 2018. The 20-year cumulative risk of dying of breast cancer was estimated using the Kaplan–Meier method.

We also estimated 10- and 20-year breast cancer-specific survival rates for women who developed invasive breast cancer; patients were followed from the date of diagnosis until the date of death from breast cancer, death from another cause or 31 December 2018.

### 4.5. Standardized Mortality Ratio

To estimate the expected number of deaths from breast cancer in the cohort if they were an average-risk population, we calculated the expected probability of death for each of the study subjects under the assumption that they did not have breast cancer at the time of the first MRI screening. To calculate the expected probability of death within a given interval for women who were initially cancer-free, we estimated the risk of breast cancer incidence in the cohort and then applied breast cancer case-fatality rates based on SEER data from the year of diagnosis until the end of the interval. Patients were censored from follow-up at the date of bilateral preventive mastectomy. For example, if a woman had a first MRI screening at age 50 and was followed until age 70, we asked: what is the probability that a cancer-free women at age 50 would develop and die of breast cancer before age 70? Accordingly, we calculated the expected probability of death from breast cancer for each study subject, according to the age at first MRI and the age at last follow up. To estimate the total expected number of deaths in the cohort, we summed the individual probabilities of death for each woman in the cohort as described above. To calculate the SMR, we compared the observed number of deaths to the expected number of deaths.

## 5. Conclusions

Among 491 previously unaffected *BRCA1/2* mutation carriers enrolled in our study of annual breast screening with MRI and mammography, after a median follow-up of 13 years, there were only four cases of breast cancer-related death. Annual screening with MRI plus mammography is associated with a 20-year probability of not dying of breast cancer of 98%. Future studies will likely demonstrate even better breast cancer-specific survival due to improved screening technology and more effective treatments.

## Figures and Tables

**Figure 1 cancers-12-03479-f001:**
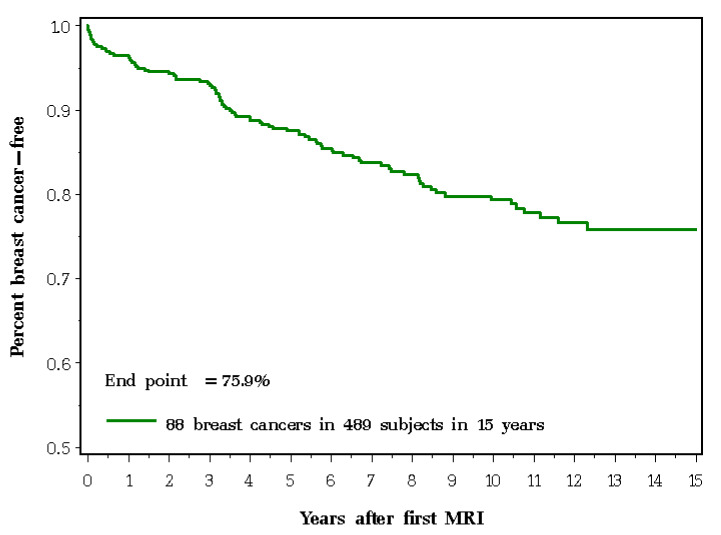
Probability of remaining breast cancer-free following first magnetic resonance imaging (MRI) screening.

**Figure 2 cancers-12-03479-f002:**
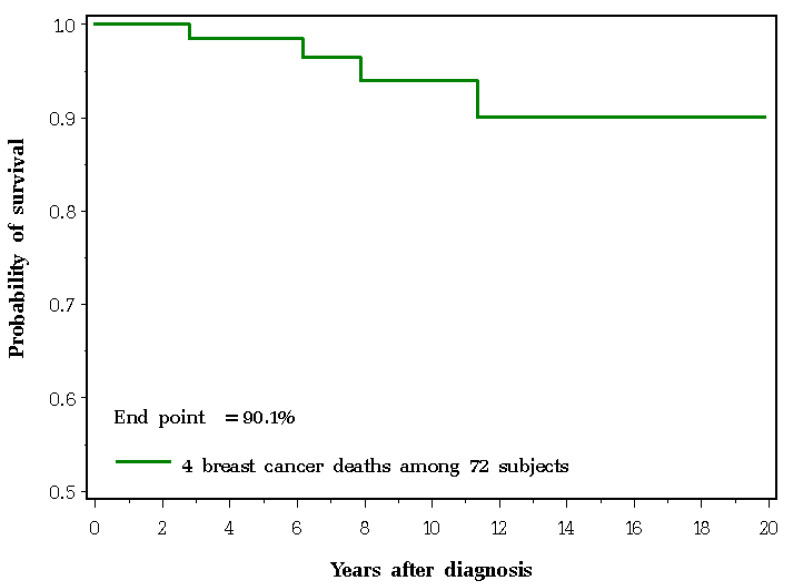
Twenty-year breast cancer-specific survival for 72 patients with invasive breast cancer.

**Table 1 cancers-12-03479-t001:** Characteristics of the 489 subjects.

Variables	Frequency or Mean Value
**Year of Birth**	1962 (1936–1986)
**Date of 1st MRI**	2005 (1997–2012)
**Age at 1st MRI**	42.8 (25–66)
**Mutation**	
BRCA1	248 (50.7%)
BRCA2	241 (49.3%)
**Preventive Mastectomy in Follow-up**	
No	394 (80.6%)
Yes	95 (19.4%)
**Oophorectomy**	
No/missing	237 (48.5%)
Yes	252 (51.5%)
**Breast Cancer**	
No	398 (81/4%)
Yes	91 (18.6%)
Year of Diagnosis	2009 (1998–2016)
**Ovarian Cancer**	
No	476 (97.3%)
Yes	13 (2.7%)
Year of diagnosis	2008 (2000–2017)
Other Cancer	
No	468 (95.7%)
Yes	21 (4.3%)
Pancreas	4
Skin	4
Cervix	3
Colorectal	2
Thyroid	2
Endometrial	1
Kidney	1
Lung	1
Lymphoma	1
Unknown	2
**Alive**	471 (96.3%)
**Dead**	18 (3.7%)
Date of death	2012 (2004–2018)
Cause of death	
Breast Cancer	4
Ovarian cancer	5
Pancreatic cancer	4
Lung cancer	1
Other/missing	4

**Table 2 cancers-12-03479-t002:** Characteristics of the 92 incident breast cancers by cancer type.

Variables	Invasive (n = 72)	DCIS (n = 19)	
**Year of birth:** mean (range)	1960 (1937–1981)	1959 (1946–1974)	
**Age at diagnosis:** mean (range)	48.6 (31–69)	48.4(35–67)	
**Tumour size (cm):** mean (range)	1.03 (0–3.0) (n = 69)	0.64 (0–2.2) (n = 14)	
**Node status**			
Negative	58 (86.6%)	6 (100%)	
Positive	9 (13.4%)	0	
Missing	5	13	
**Stage**			
1	49 (74.2%)	n/a	
2 or higher	17 (25.8%)		
**Estrogen Receptor status**			
Positive	41 (59.4%)	6 (66.7%)	
Negative	28 (40.6%)	3 (33.3%)	
Missing	3	10	
**Chemotherapy**			
No	32 (47.8%)	14 (100%)	
Yes	35 (52.2%)	0	
Missing	5	5	

**Table 3 cancers-12-03479-t003:** Subjects who died of breast cancer.

Study ID	Gene	Yr of Birth	Age at 1st MRI	Age at Diagnosis	Means of Detection	Size (cm)	Node Status	ER	Chemo	HT	Age at Recurrence	Age of Death
296	*BRCA2*	1942	59	63	MRI	0.7	-	+	No	No	70	74
424	*BRCA1*	1974	30	33	Self	1.5	+	-	Yes	n/a	35	37
455	*BRCA1*	1955	48	48	MRI	3.0	+	+	Yes	Yes	54	55
566	*BRCA1*	1965	41	45	MRI	0.11	-	-	No	n/a	50	53

HT—hormone therapy.

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
