# Peer review of "Breast Cancer Mortality among Women with a BRCA1 or BRCA2 Mutation in a Magnetic Resonance Imaging Plus Mammography Screening Program"

_cancers, 2020, doi:10.3390/cancers12113479_

Round 1

Reviewer 1 Report

This contribution is an update of the paper "Long-term results of screening with magnetic resonance imaging in women with BRCA mutations" BJC 2012, 107, 24-30 and reports about a long term prospective observational study of BRCA mutation carriers started in 1997 in the province of Ontario, Canada. The paper gives survival data of the women involved in the study. Just 4 breast cancer related deaths were found after a median follow-up of 13 years. The authors conclude, that an annual screening with MRI and mammography for this high risk is associated with a 20 year probability of 98.2% not to die of breast cancer.

One must keep in mind that the cohort size is not very large and the follow-up period is still rather short. Nevertheless the MRI/mammography surveillance seems to be a reasonable alternative to prophylactic mastectomy, which gives hope for the carriers of these mutations.

Reading this contribution requires prior study of the above mentioned BJC paper, which reduces the quality of the presentation. The chapters 4.4 and 4.5 are hard to follow and would benefit from some more detail.

Author Response

Reviewer 1

This contribution is an update of the paper "Long-term results of screening with magnetic resonance imaging in women with BRCA mutations" BJC 2012, 107, 24-30 and reports about a long term prospective observational study of BRCA mutation carriers started in 1997 in the province of Ontario, Canada. The paper gives survival data of the women involved in the study. Just 4 breast cancer related deaths were found after a median follow-up of 13 years. The authors conclude that an annual screening with MRI and mammography for this high risk is associated with a 20 year probability of 98.2% not to die of breast cancer.

One must keep in mind that the cohort size is not very large and the follow-up period is still rather short. Nevertheless the MRI/mammography surveillance seems to be a reasonable alternative to prophylactic mastectomy, which gives hope for the carriers of these mutations.

Reading this contribution requires prior study of the above mentioned BJC paper, which reduces the quality of the presentation. The chapters 4.4 and 4.5 are hard to follow and would benefit from some more detail.

The previous paper in the BJC described incidence and stage in the cohort. This is the first paper to report on cancer specific mortality.

We have expanded and clarified sections 4.4 and 4.5 (statistical methods)

Reviewer 2 Report

Warner et al. report on breast cancer mortality among women with a BRCA1 2 or BRCA2 mutation in a magnetic resonance imaging screening program. They reported the long-term breast-cancer specific survival and causes of death for women in the cohort of 491 previously unaffected BRCA1/2 mutation carriers aged 25 to 65 years, who were screened with annual MRI plus mammography between 1997 and June 2011. There were 92 women diagnosed with breast cancer with median follow-up 12.7 (range 8.5 to 22.5) years. Four deaths from breast cancer were observed, compared to 2.0 deaths expected (SMR 2.0, p = 0.14). For the 491 women in the study, the probability of not dying of breast cancer at 20 years from the date of first MRI was 98.2%. However, authors did not report clearly when 124 preventive bilateral mastectomies were done. Furthermore, they did not report clearly the risk of diagnosis of breast cancer and the cancer specific survival in those patients, who had active surveillance only. My view is that major revision should be done before the publication in Cancers.

Major remarks

Figures 1 and 2 are missing

The title should include “plus mammography”

Please provide data about timing of bilateral prophylactic mastectomy (age of patients, year, how many in patients without cancer, how many contralateral prophylactic mastectomies in patients with cancer)

Please provide data about timing of prophylactic oophorectomy (age of patients, year)

Table 1 should be reorganized (all patients versus patients with surveillance only versus patients with prophylactic mastectomy)

Lines 94-95: Follow up of patients is too short to report 20-year cumulative incidence of breast cancer.

Lines 123-124: Was the ten-year survival and the twenty-year survival of patients with invasive breast cancer calculated from the time of diagnosis of breast cancer? Please, specify it in the methods section.

Line 124-125: The sentence “the actuarial risk of not dying of breast cancer at 20 years from the date of the first MRI was 98.2%” is misleading because more than half of patients had prophylactic oophorectomy and 27% had bilateral prophylactic mastectomy. Please, state results for all patients and separately for those who were on surveillance only and for those who had bilateral prophylactic mastectomy.

Line 148-150: In women struggling with the decision of breast screening vs. mastectomy, another critical question is the possible treatment of cancer (chemotherapy, hormonal treatment, external beam irradiation, their side effects and late complications).

Line 163: One possible explanation is also that this study did not report separately patients who were only on surveillance and those who had prophylactic surgery.

Line 211: Reference 28 state: “It is plausible that oophorectomy may reduce breast cancer risk in BRCA2 mutation carriers but not in BRCA1 mutation carriers”!

Line 286-287: “Despite the limitations of our study we believe we have not overestimated the effectiveness of breast screening”. Please delete this sentence. The authors should state results for all patients, for those who were on surveillance only and for those who had bilateral prophylactic mastectomy.

Lines 290-301: This two paragraphs should be moved to the discussion section.

Minor remarks

Line 49: For the 1 in 400 women??

Table 1: Line “Year of breast cancer” – How many patients had breast cancer in 2017, 2018 and 2019?

Table 1. Please provide data about types of other cancer for 29 patients.

Line 130: There were 11 additional deaths from cancer. In table 2 there are only 10.

Line 134: 13 years is not 12.7 years

Line 181-182: number of MRI and/or mammography investigations

Line 250: details of MRI and/or mammography examinations

Author Response

Reviewer 2

Warner et al. report on breast cancer mortality among women with a BRCA1 or BRCA2 mutation in a magnetic resonance imaging screening program. They reported the long-term breast-cancer specific survival and causes of death for women in the cohort of 491 previously unaffected BRCA1/2 mutation carriers aged 25 to 65 years, who were screened with annual MRI plus mammography between 1997 and June 2011. There were 92 women diagnosed with breast cancer with median follow-up 12.7 (range 8.5 to 22.5) years. Four deaths from breast cancer were observed, compared to 2.0 deaths expected (SMR 2.0, p = 0.14). For the 491 women in the study, the probability of not dying of breast cancer at 20 years from the date of first MRI was 98.2%. However, authors did not report clearly when 124 preventive bilateral mastectomies were done. Furthermore, they did not report clearly the risk of diagnosis of breast cancer and the cancer specific survival in those patients, who had active surveillance only. My view is that major revision should be done before the publication in Cancers.

We have added the requested detail regarding the patients who had a prophylactic mastectomy throughout the paper. Note that 95 of the patients had a prophylactic mastectomy. This has been corrected.   There were four breast cancers diagnosed at prophylactic mastectomy and none after prophylactic mastectomy. There were no deaths from breast cancer in the 95 women who had a preventive mastectomy.   We have added these details throughout the paper including in the Simple Summary and the Abstract.   Also see lines 83 to 90

Major remarks

Figures 1 and 2 are missing.

They are now included in the manuscript after the paragraph where they appear in the text

The title should include “plus mammography”

change made.

Please provide data about timing of bilateral prophylactic mastectomy (age of patients, year, how many in patients without cancer, how many contralateral prophylactic mastectomies in patients with cancer). Please provide data about timing of prophylactic oophorectomy (age of patients, year)

The data is added lines 83-90 and in table 1.

Table 1 should be reorganized (all patients versus patients with surveillance only versus patients with prophylactic mastectomy).

We have included the details regarding the prophylactic mastectomy patients in the text. The study was not designed to compare patients with mastectomy to patients with MRI and this will be to topic of a future paper. Note that in this paper all subjects had an MRI including those with a prophylactic mastectomy.

Lines 94-95: Follow up of patients is too short to report 20-year cumulative incidence of breast cancer.

We now include the ten and 15 year cumulative incidence of breast cancer line 111. We have revised figure 1 to highlight 15 year survival.

Lines 123-124: Was the ten-year survival and the twenty-year survival of patients with invasive breast cancer calculated from the time of diagnosis of breast cancer? Please, specify it in the methods section.

Yes this has been clarified line 321.

Line 124-125: The sentence “the actuarial risk of not dying of breast cancer at 20 years from the date of the first MRI was 98.2%” is misleading because more than half of patients had prophylactic oophorectomy and 27% had bilateral prophylactic mastectomy. Please, state results for all patients and separately for those who were on surveillance only and for those who had bilateral prophylactic mastectomy.

This is correct because the cohort was censored for follow up at the date of bilateral mastectomy. Therefore the risk is based only on women with two breasts intact. We state this on line 299. We have corrected the data on the number of women who had bilateral preventive mastectomy. We have also added a line reporting the survival of the 394 women who did not have a bilateral mastectomy at any point (lines 154, 155)

Line 148-150: In women struggling with the decision of breast screening vs. mastectomy, another critical question is the possible treatment of cancer (chemotherapy, hormonal treatment, external beam irradiation, their side effects and late complications)

We have added a comment on lines 180, 181…. and the possibility of a breast cancer diagnosis and treatment (radiation, chemotherapy, and/or hormonal therapy) with its side effects and complications.

Line 163: One possible explanation is also that this study did not report separately patients who were only on surveillance and those who had prophylactic surgery.

We have now clearly identified the patients. All outcomes were censored at bilateral preventive mastectomy. We have also added data on the survival of women who did not have a bilateral preventive mastectomy

Line 211: Reference 28 state: “It is plausible that oophorectomy may reduce breast

cancer risk in BRCA2 mutation carriers but not in BRCA1 mutation carriers”!

This sentence has been revised to read: “although risk reduction was not found in BRCA1 mutation carriers or in BRCA2 mutation carriers who had undergone this surgery within the previous five years in a more recent study.”

Line 286-287: “Despite the limitations of our study we believe we have not

overestimated the effectiveness of breast screening”. Please delete this sentence. The authors should state results for all patients, for those who were on surveillance only and for those who had bilateral prophylactic mastectomy.

This sentence has been deleted.

Lines 290-301: This two paragraphs should be moved to the discussion section.

This has been moved near to appear just before the discussion of other cancers and the references rearranged accordingly.

Minor remarks

Line 49: For the 1 in 400 women

 The sentence has been revised for clarity

Table 1: Line “Year of breast cancer” – How many patients had breast cancer in 2017, 2018 and 2019?

There were no patients diagnosed with breast cancer after 2016. We have now corrected this.

Table 1. Please provide data about types of other cancer for 29 patients.

There were 21 other cancers in the follow up period. This has been corrected. The cancers are listed in table 1.

Line 130: There were 11 additional deaths from cancer. In table 2 there are only 10.

There were ten additional cancer deaths, this has been corrected. Line 163

Line 134: 13 years is not 12.7 years

The change has been made

Line 181-182: number of MRI and/or mammography investigation

 The change has been made

Line 250: details of MRI and/or mammography examinations

The change has been made

Reviewer 3 Report

This article reported the breast cancer mortality of the women with a BRCA1 or BRCA2 mutation followed by MRI screening program. They concluded that the annual screening with MRI plus mammography is a safe option for women with BRCA mutation who decline or defer risk-reducing mastectomy because of the low rate of breast cancer death (2%). This clinical question is very interesting and important. However, I don’t think that these results were enough for the authors’ conclusion.

  1. The authors should indicate the standard or control data to evaluate the value of this screening system. There were 60 women diagnosed breast cancer, and 13 patients were not diagnosed by MRI screening. There were 4 patients died of breast cancer, and 2 of them were diagnosed by MRI screening (one was by self and one was by first MRI.) How did the authors think the meaning of these results?

  1. The authors included analyzed together both with and without preventive mastectomy and oophorectomy. These surgeries influenced to the prognosis. I think that the authors should exclude these patients received preventive surgery. Moreover, the patients diagnosed breast cancer by the first MRI should be excluded for main analysis.

  1. The breast cancer type with BRCA1 or BRCA2 mutation were different. There are previous articles reported that the breast cancer of women with BRCA2 were relatively diagnosed by MMG with microcalcifications. How do you think that the difference or importance of BRCA mutations for MRI screening systems.

Author Response

This article reported the breast cancer mortality of the women with a BRCA1 or BRCA2 mutation followed by MRI screening program. They concluded that the annual screening with MRI plus mammography is a safe option for women with BRCA mutation who decline or defer risk-reducing mastectomy because of the low rate of breast cancer death (2%). This clinical question is very interesting and important. However, I don’t think that these results were enough for the authors’ conclusion.

  1. The authors should indicate the standard or control data to evaluate the value of this screening system. There were 60 women diagnosed breast cancer, and 13 patients were not diagnosed by MRI screening. There were 4 patients died of breast cancer, and 2 of them were diagnosed by MRI screening (one was by self and one was by first MRI.) How did the authors think the meaning of these results?

 The sensitivity of screening breast MRI alone performed annually is only about 90% (Reference 12) and rises to 95% when combined with mammography. The denominator for that statistic is cancers detected either clinically (ie palpable cancers) or by screening.

Of the 13 cancers we mentioned that were not detected by MRI, 4 were detected at prophylactic mastectomy and would likely have been detected while still subclinical at the next round of breast screening. Of the remaining 9 cancers 3 were palpable by a formal clinical breast examination done at the time of the screening MRI as well as by the screening MRI itself (but had not presented earlier as interval cancers). Six cancers not detected by MRI is what one would expect.

We acknowledge that not every cancer detected by screening MRI will be curable. However, the finding in our paper is that the great majority of women in a population of BRCA mutation carriers screened annually with MRI and mammography do not die of their breast cancer in a 20 year follow up period. . The breast cancer specific survival of our cohort is not significantly higher than that in the non-carrier (SEER population)

2. The authors included analyzed together both with and without preventive mastectomy and oophorectomy. These surgeries influenced to the prognosis. I think that the authors should exclude these patients received preventive surgery. Moreover, the patients diagnosed breast cancer by the first MRI should be excluded for main analysis. We excluded patients who had a bilateral mastectomy before the first MRI. We did not exclude patients who had a mastectomy after the start of the study from the analysis but we censored the events at the time of preventive mastectomy. It would be inappropriate to exclude them because they were enrolled in the original study and it is necessary to follow the entire cohort for the relevant outcomes.  

3.The breast cancer type with BRCA1 or BRCA2 mutation were different. There are previous articles reported that the breast cancer of women with BRCA2 were relatively diagnosed by MMG with microcalcifications. How do you think that the difference or importance of BRCA mutations for MRI screening systems? A:

Several authors have found relatively low sensitivity of mammography for BRCA1 mutations carriers (particularly those under aged 50) with higher sensitivity of mammography for BRCA2. However, even for BRCA2 mutation carriers, the sensitivity of screening mammography is much lower than the sensitivity of MRI (see reference 12). In the first phase of our study (reference 14) the sensitivity of mammography for BRCA2 mutation carriers was 20% compared to 80% for MRI. All guidelines (eg. NCCN, ESMO) currently recommend similar screening protocols (with annual MRI plus mammography) for BRCA1 and BRCA2 mutation carriers. The focus of the current paper is on mortality rather than sensitivity and this outcome has not been reported before in our cohort

Round 2

Reviewer 2 Report

Warner et al. report on breast cancer mortality among women with a BRCA1 2 or BRCA2 mutation in a magnetic resonance imaging screening program. They reported the long-term breast-cancer specific survival and causes of death for women in the cohort of 491 previously unaffected BRCA1/2 mutation carriers aged 25 to 65 years, who were screened with annual MRI plus mammography between 1997 and June 2011. There were 92 women diagnosed with breast cancer with median follow-up 12.7 (range 8.5 to 22.5) years. Of the 489 women who were in the cohort, 95 had a bilateral preventive mastectomy in the follow-up period. All four of the deaths were among the 394 women who did not have a bilateral mastectomy. My view is that paper can be published in Cancers.

Reviewer 3 Report

I think that the authors replied my questions and added the sentences. This article has enough value to be accepted to this journal.